# Parameter-Efficient VLMs for Gastrointestinal Endoscopy: Medical Image Generation and Clinical Visual Question Answering

Ojonugwa Oluwafemi Ejiga Peter[1], Frederick Akor Ejiga[2], Fahmi Khalifa[3], and Md Mahmudur Rahman[1,*]

*Abstract*—The major limitations of the gastrointestinal (GI) endoscopy AI systems arise from a shortage of annotated data, strict privacy policies, and significant bottlenecks in conventional model fine-tuning. Such limitations impede the successful application of sophisticated AI models in clinical practice, particularly affecting the reliability and scalability of diagnosis. In this paper, we present a dual-pipeline, PEFT model that addresses two fundamental problems, including medical Visual Question Answering (VQA) and the generation of privacy-preserving synthetic data. For clinical VQA, we adopt the Florence-2 vision-language model. Leveraging PEFT enhances model interpretability while substantially reducing the computational cost of training. Simultaneously, we employ Low-Rank Adaptation (LoRA) in the context of Stable Diffusion 2.1 to generate high-quality GI images that enhance training databases without violating patient privacy. This research utilized the Kvasir-VQA dataset. Our Florence-2 VQA model scored ROUGE-1 0.92, ROUGE-L 0.91, and improvements in BLEU scores of 0.08–0.24. Fine-tuning on private datasets consistently showed better results than on public datasets. The rank-4 LoRA synthesis achieved optimal performance with a fidelity score of 0.290, an agreement score of 0.730, and a Frechet BiomedCLIP Distance (FBD) of 1450, reducing computational costs by almost 90%. This framework improves the clinical potential of AI in GI endoscopy. Compared to FLUX, MSDM, and Kandinsky 2.2, our model demonstrates superior FBD and strong semantic alignment. While others lead in Fidelity or Agreement, our lower FBD indicates better image-text coherence. These results establish our approach as a robust solution for enhancing VQA and synthetic data generation in clinical AI.

*Index Terms*—Biomedical imaging, Gastrointestinal endoscopy, Low-Rank Adaptation (LoRA), Medical image generation, Medical visual question answering, Parameter-efficient fine-tuning (PEFT), Data augmentation, Stable Diffusion, Synthetic data generation, Transformer architectures, Vision-language models, Florence-2, Image generation, Cross-attention mechanisms, Polyp detection, Synthetic Medical Imaging,

## I. INTRODUCTION

Gastrointestinal (GI) endoscopy is the gold standard for diagnosing and treating digestive tract diseases, offering direct visual access within the body and enabling important procedures such as tissue biopsies and the removal of growths like intestinal polyps [1]. Nevertheless, deploying advanced artificial intelligence (AI) systems in GI endoscopy faces significant barriers, including the scarcity of annotated image datasets, strict patient privacy requirements, and costly expert labeling efforts [2]. Current AI solutions often require large, annotated datasets and intensive computation, yet still suffer from bias and poor generalization to rare pathologies [3]. Synthetic data offers a promising solution to privacy concerns and dataset limitations in medical imaging. Diffusion-based generative models and Vision-Language Models (VLMs) can produce clinically relevant, high-fidelity images for diagnostic use [2]. By combining PEFT with synthetic data and Visual Question Answering (VQA), we enable more accurate and resource-efficient medical AI solutions. This paper proposes a dual-pipeline framework: optimizing Florence-2 for gastrointestinal VQA and using LoRA-enhanced Stable Diffusion to generate privacy-preserving GI images for dataset expansion. The primary contributions of this work are as follows: (1) implementation of PEFT techniques for the Florence-2 model on the Kvasir-VQA dataset, accompanied by an in-depth performance analysis; (2) design of a robust synthetic image generation framework utilizing diffusion models augmented with advanced prompt engineering methods; and (3) thorough evaluation of both methodologies in the context of gastrointestinal clinical diagnostics, supported by detailed ablation studies.

## II. RELATED WORKS

### A. *Parameter-Efficient Fine-Tuning (PEFT)*

PEFT enables the adaptation of large pre-trained AI models to medical tasks by freezing most weights and updating only a small number of trainable parameters, reducing computation and avoiding catastrophic forgetting [4]. A leading PEFT method is Low-Rank Adaptation (LoRA) [4], which inserts low-rank trainable matrices into transformer attention blocks. LoRA-personalized models, when used for lung nodule classification, match traditional performance while reducing trainable parameters by 89.9% and training time by 36.5% [5]. Integrating LoRA with generative models like Stable Diffusion and DreamBooth enables efficient, high-fidelity medical image

This work was supported by the National Science Foundation (NSF) grant (ID. 2131307) "CISE-MSI: DP: IIS: III: Deep Learning-Based Automated Concept and Caption Generation of Medical Images Towards Developing an Effective Decision Support."

[1] Computer Science Department, Morgan State University, Baltimore, USA.
[2] International Organization for Migration (IOM), Geneva, Switzerland.
[3] Electrical & Computer Engineering Department, Morgan State University, Baltimore, USA.
[*] Corresponding author: md.rahman@morgan.edu.

generation [3]. DreamBooth is particularly useful for fine-tuning rare conditions [3]. Prompt tuning combines visual and textual inputs to capture detailed clinical features [6], while spatial cues like scribbles and bounding boxes guide model attention in Medical VQA tasks [7]. Remarkably, the 11B-parameter DermatoLlama was fine-tuned with LoRA on a single RTX 4090, proving PEFT's scalability in constrained settings [8], [9].

### B. Synthetic Data Generation and Privacy Preservation

Synthetic data generation addresses privacy, imbalance, and data scarcity in medical imaging by maintaining patient privacy while enabling global data sharing within institutions [2]. Diffusion models, especially Stable Diffusion, transform the generation of synthetic medical images using text-to-image synthesis and achieve low Fréchet Inception Distance (FID) scores (0.064-0.099) and high Inception Score (IS) values (average 2.327) during the MedVQA-GI challenge [10]. Improved StarGAN models produce endoscopic images in varied viewing perspectives, and improve classification accuracy on the Kvasir database by 0.704 percentage points [11].

### C. Medical Visual Question Answering and Hallucination Mitigation

Medical VQA allows AI systems to interpret endoscopic images and respond accurately to clinical queries. [12]–[14]. Nevertheless, GI endoscopy VQA faces serious challenges, including the lack of data and annotation difficulty, and hallucinations that produce seemingly reasonable yet incorrect information [15]–[17]. One GI dataset shows no more than 30.39 percent completely accurate VLM-elicited responses in a research study [17]. Developed hallucination-aware fine-tuning approaches consistently outperform conventional methods, achieving 90.89 percent Question Answering Accuracy Score (QAAS) for LLaVA-1.6-7B [17]. MedPLIB promotes pixel-level comprehension using visual cues for clinical assessment [18], whereas multi-task fine-tuning integrates detection, localization, and counting into endoscopic images, reflecting clinical practices [19].

### D. Performance Evaluations and Specialized Datasets

Specialized datasets drive GI endoscopy image research: Gut-VLM consists of expert-labeled hallucination correction tags [17]; EndoBench offers full benchmarking of 4 endoscopic scenarios, 12 clinical tasks, and 6,832 validated VQA pairs across 21 datasets [16]; MedMultiPoints addresses multi-task medical image understanding [19]; HyperKvasir demonstrates GPT-4 Vision's ability to perform few-shot classification of Boston (Bowel Preparation Scale) [20]–[22]. The effectiveness of PEFT-enabled systems is well demonstrated through comprehensive evaluations. Automated polyp segmentation protocols use Stable Diffusion-augmented synthetic data combined with Faster R-CNN for object localization and advanced segmentation characteristics such as Segment Anything Model (SAM), U-Net, LinkNet, and MANet [22]. Models trained on data-augmented datasets often demonstrate superior generalization performance when evaluated on original data. Transformers (DeiT, ViT) showed 93% on an augmented data set, and EfficientNet showed 97 percent on baseline datasets [3]. U-Net achieved 0.65 IoU and 0.78 Dice coefficient in polyp segmentation [22].

### E. Critical Gaps and Future Directions

AI models often lack generalizability and robustness to varying data distributions, requiring large, high-quality datasets across demographics, equipment, and diseases [2], [23], [24]. Augmenting real data with synthetic counterparts enhances performance, proving synthetic data's value as a supplement, not a replacement [3], [25]. Future directions include improving robustness and reducing hallucinations through fine-tuning and human-in-the-loop validation [10], expanding synthetic datasets to better reflect GI diversity [2], and establishing ethical, regulatory frameworks for clinical deployment [12]. Integrating PEFT, VLMs, synthetic generation, and VQA promises to alleviate data scarcity and privacy issues while advancing reliable, explainable, and cost-effective AI-driven GI diagnostics [2], [23].

## III. METHODOLOGY

The study employs a dual-pipeline design approach covering two aspects of Medical AI: visual question answering for clinical decision making and synthetic data generation, supported by synthetic image synthesis. The methodology employs PEFT methods to adapt state-of-the-art VLMs and diffusion-based generative systems specifically to gastrointestinal endoscopies. This paper adopts a dual-pipeline architecture for gastrointestinal endoscopic analysis: (a) A parameter-efficient vision-language VQA pipeline based on Florence-2 and (b) A LoRA-enhanced Stable Diffusion pipeline for synthetic image generation

### A. Data Preparation

The Kvasir-VQA dataset (6,500 images) [24] supports both tasks by providing diverse VQA pairs across gastrointestinal pathologies and medical instruments. Each image includes multiple question-answer pairs, covering yes/no, counting, and object localization questions. Images are resized to $512 \times 512$ RGB to preserve clinical detail. An 80/20 train-validation split $(5,200/1,300$ images) ensures consistent evaluation across both pipelines. Medical questions are prefixed with the domain-specific token `<MedVQA>` to activate relevant representations. Synthetic captions are enhanced with descriptors such as "clinical colonoscopy image" to guide the generation process toward medically accurate outputs.

### B. System Architecture

The dual-pipeline architecture facilitates the parallel development of diagnostic capabilities and privacy-preserving data augmentation for gastrointestinal endoscopy using PEFT models. The VQA pipeline freezes the vision encoder and adapts language components, while the synthetic generation pipeline applies LoRA to align diffusion model parameters

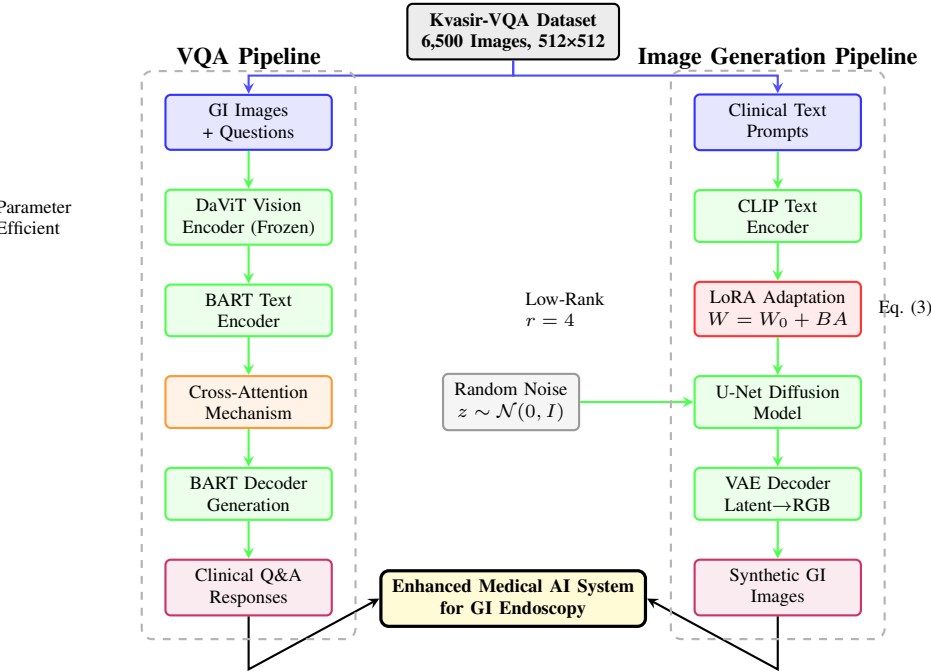

Fig. 1. Dual-pipeline architecture for PEFT gastrointestinal endoscopy AI system. Left: Florence-2-based VQA pipeline with frozen vision encoder and PEFT. Right: LoRA-enhanced Stable Diffusion pipeline for synthetic image generation. Both pipelines converge to create enhanced medical AI capabilities with reduced computational requirements.

with the medical domain. Both pipelines employ PEFT training methods tailored to clinical contexts. Figure 1 illustrates the integrated architecture supporting self-sufficient, domain-adaptive training and generation for enhanced medical AI performance.

### C. Training Algorithm

---

**Algorithm 1** Dual-Pipeline PEFT Training

---

1: **Input:** VQA dataset $\mathcal{D}_V$, synthetic prompts $\mathcal{P}$, epochs $E_V, E_S$
2: Initialize Florence-2 and Stable Diffusion models
3: **for** epoch = 1 to $E_V$ **do**
4:     **for** batch $(I, q, a)$ in $\mathcal{D}_V$ **do**
5:         Extract $K, V = \text{Enc}_{vis}(I)$ and $Q = \text{Enc}_{txt}(q)$
6:         Generate $\hat{a} = \text{Dec}(Q, K, V)$
7:         Compute $\mathcal{L}_{VQA}$ using Equation (2)
8:         Update VQA adapter parameters via AdamW
9:     **end for**
10: **end for**
11: **for** epoch = 1 to $E_S$ **do**
12:     **for** prompt $p$ in $\mathcal{P}$ **do**
13:         Sample latent $z \sim \mathcal{N}(0, I)$
14:         Generate synthetic $x = \text{SD}(z, p)$
15:         Compute FBD against reference distribution
16:         Update LoRA weights $(A, B)$ via AdamW using Equation (3)
17:     **end for**
18: **end for**
19: **Output:** Fine-tuned VQA model, LoRA adapters

---

### D. VQA Pipeline

The Florence-2 base model comprises a frozen DaViT (Data-efficient Vision Transformer) vision encoder and a BART-style decoder, creating a unified sequence-to-sequence framework supporting a wide range of vision-language tasks.

The vision encoder is a hierarchical transformer that goes through gastrointestinal images and produces visual features at multiple scales, which captures information about detailed anatomy and broader context simultaneously. Domain-specific features are activated by prefixing clinical questions with the token `<MedVQA>`. The cross-attention mechanism enables dynamic interaction between visual features and textual queries, which is defined as:

$$\text{Attention}(Q, K, V) = \text{softmax}\left(\frac{QK^T}{\sqrt{d_k}}\right) V, \qquad (1)$$

where $Q \in \mathbb{R}^{T \times d_k}$ are textual query embeddings, and $K, V \in \mathbb{R}^{N \times d_k}$ are visual key/value embeddings. This mechanism enables the model to attend to specific image regions based on question content, supporting spatial reasoning, numerical counting, and categorical classification required for medical VQA tasks.

Answer generation is supervised via token-level cross-entropy loss:

$$\mathcal{L}_{\text{VQA}} = -\sum_{t=1}^{T} \log p_\theta(a_t \mid a_{<t}, I, q), \qquad (2)$$

with $a_t$ the ground-truth token at position $t$, conditioned on image $I$ and question $q$, where $\theta$ denotes model parameters. This autoregressive loss formulation supports diverse answer formats required across question categories. Optimization uses AdamW (learning rate $2 \times 10^{-5}$, weight decay 0.01), cosine scheduling (200 warmup steps), 10 epochs, batch size 2 (gradient accumulation 8), and mixed-precision (FP16).

## E. Synthetic Image Generation Pipeline

The synthetic image generation pipeline employs Stable Diffusion 2.1 as the foundational text-to-image model. Operating within a compressed latent space, Stable Diffusion 2.1 achieves high-resolution outputs while maintaining computational efficiency—an essential feature for medical imaging tasks with limited resources. To enable effective adaptation to the gastrointestinal domain, Low-Rank Adaptation (LoRA) is integrated into the attention layers of the diffusion model. LoRA modifies pre-trained attention weight matrices without full retraining, enabling PEFT. Specifically, the updated attention weights are defined as:

$$W = W\_0 + \Delta W, \quad \Delta W = B, A, \tag{3}$$

where $W_0 \in \mathbb{R}^{d \times k}$ is the frozen base matrix, and $B \in \mathbb{R}^{d \times r}, A \in \mathbb{R}^{r \times k}$ are the trainable components with $r \ll \min(d, k)$. This decomposition reduces tunable parameters from dkdk $dk$ to $r(d + k)$, significantly reducing overfitting risk on small medical datasets. Empirically, rank-4 provides an optimal balance between efficiency and representational power. LoRA fine-tuning is performed using AdamW (learning rate: $1 \times 10^{-4}$), cosine scheduling with 500 warm-up steps, 10 training epochs, a batch size of 4, and gradient accumulation of 2. Evaluation metrics, including FBD and domain-specific fidelity scores, are computed per epoch to ensure clinical realism and image quality.

## IV. RESULTS

This section presents a comprehensive assessment of our parameter-efficient VLMs for gastrointestinal endoscopy, demonstrating the effectiveness of clinical visual question answering and medical image generation. Our PEFT-based Florence-2 architecture demonstrates strong performance across diverse medical VQA tasks. Results in Figure 2 show ROUGE-1 scores improving from 0.61 to 0.92 in the best configuration, while ROUGE-L achieved 0.91, representing better recall and sequence alignment with ground-truth medical answers. BLEU scores showed significant improvement (increases of 0.08 to 0.24), indicating improved n-gram accuracy for medical terminology. METEOR scores remained steady at 0.31-0.50, demonstrating consistent semantic similarity during PEFT adaptation. Results consistently show that fine-tuning on private datasets outperformed public datasets, with ROUGE-1 of 0.910 versus 0.792 for approaches using public datasets, and comparable results in the ROUGE-L (0.905 vs 0.785) and BLEU (0.210 vs 0.177) scores.

Systematic evaluation reveals consistent and robust performance trends across various testing configurations. These findings confirm the effectiveness of our PEFT strategy in maintaining high VQA accuracy across diverse clinical questions and image scenarios. Performance consistency demonstrates the model's generalization ability across categories, question types, and anatomical contexts, validating its reliability for medical imaging applications. Furthermore, the model maintains stable performance across varied question

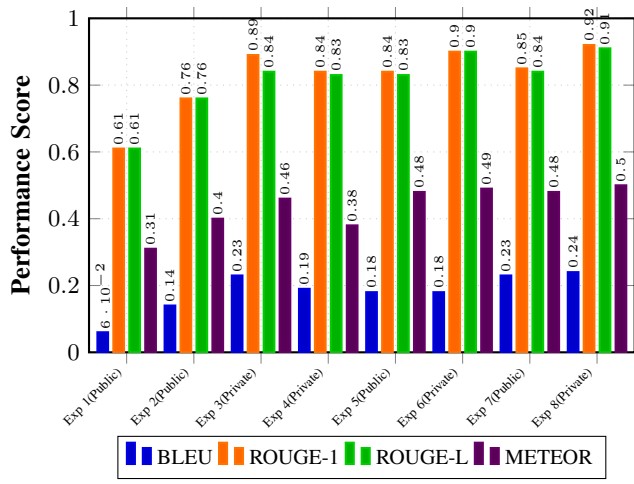

Fig. 2. Performance comparison of VQA metrics across eight experiments using public and private datasets. Higher scores indicate better performance across all metrics (BLEU, ROUGE-1, ROUGE-L, and METEOR).

formats and in the presence of visual ambiguity, reinforcing its robustness. These outcomes collectively highlight the architectural strengths of the dual-pipeline training process. As shown in Figure 2, results demonstrate high alignment with expert-level reasoning across experimental conditions. Beyond quantitative metrics, qualitative analysis provides deeper insights into model interpretability and clinical applicability. The model demonstrates strong spatial reasoning and anatomical localization capabilities. It accurately identifies abnormality locations in gastrointestinal endoscopic images, labeling them with anatomical references such as "central," "lower-central," and "upper-right." These semantically grounded outputs align with clinical expectations and enhance diagnostic reliability, as illustrated in Figure 3.

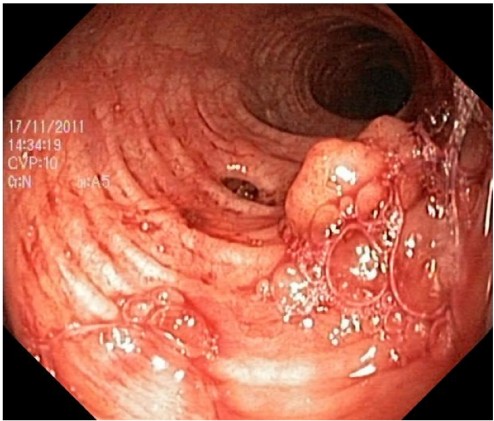

**Question:** How many polyps are in the image?
**Ground Truth:** 1.
**Answer:** 1.

Fig. 3. Visual Question Answering (VQA) example for gastrointestinal endoscopy.

LoRA-enhanced Stable Diffusion 2.1 models provide a practical approach to addressing medical image scarcity through high-quality synthesis. Our low-rank #4 Low-Rank Adaptation setup does not overfit but retains anatomical detail. Analysis in Figures 4 and 5 shows that Fidelity, Agreement, and FBD metrics across three experimental trials exhibit distinct optimization behaviors.

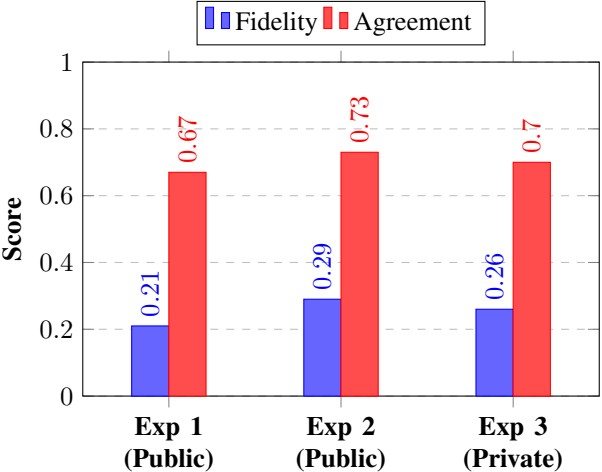

Fig. 4. Fidelity and Agreement scores across three experiments using public and private datasets.

Trial 2 achieved optimal performance with fidelity of 0.290, agreement of 0.730, and FBD of 1450, providing an ideal trade-off between structural accuracy and semantic alignment. Unexpectedly, public dataset training outperformed private datasets for synthetic generation, unlike VQA results. Public data training (trials 1-2) yielded better FBD values (2022 and 1450) compared to private data (trial 3: 1539), although competitive fidelity and agreement values were obtained.

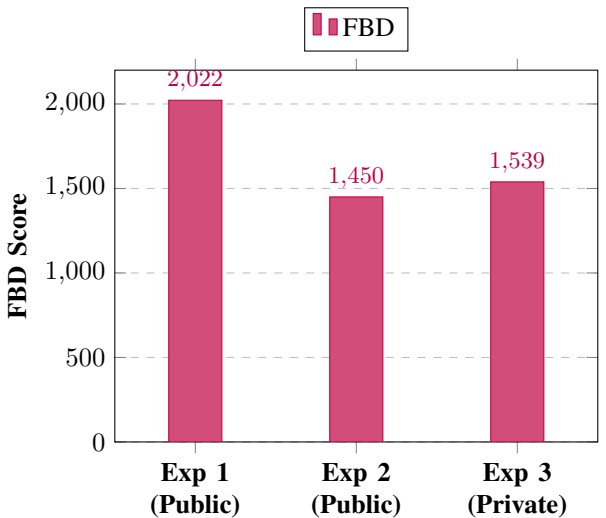

Fig. 5. Frechet BiomedCLIP Distance (FBD) scores. Lower values indicate better semantic alignment between generated and real medical images.

Synthetic images exhibit clinically realistic features, as shown in Figure 6, including appropriate endoscopic lighting, detailed mucosal texture, and accurate polyp appearances. Generated images maintain diagnostically relevant characteristics necessary for training medical applications, and expert validation confirmed the clinical utility of this data augmentation approach. The PEFT approach significantly reduces computational requirements compared to full fine-tuning. Using mixed precision and gradient checkpointing, we reduced training time to approximately 4–5 hours on standard GPU hardware. Memory optimization enables implementation on standard hospital infrastructure without requiring specialized computational resources.

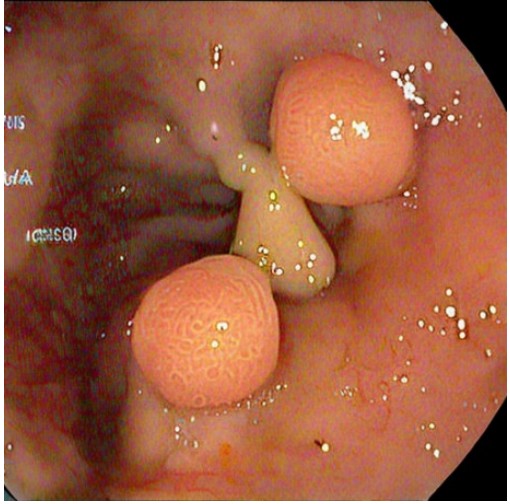

**Prompt:** Generate a GI image with a polyp.

Fig. 6. High-fidelity GI images generated using clinical prompts.

Error analysis reveals fundamental challenges in numerical counting tasks within complex multi-lesion scenarios. Quantification precision occasionally suffers when multiple pathological features appear simultaneously. However, these limitations do not significantly impact overall clinical utility, particularly for diagnostic assistance and education. Although formal expert Likert-scale validation was not conducted, all synthetic images were manually inspected for anatomical plausibility, mucosal structure integrity, and endoscopic realism. This internal validation was used to ensure basic clinical believability, though we acknowledge the need for structured clinical scoring as a future step.

Key design choices were validated through systematic component analysis. LoRA rank experiments confirm that rank-4 adaptation provides the optimal trade-off between parameter capacity and overfitting prevention. Higher ranks yielded diminishing returns while increasing computational overhead. The frozen vision encoder approach achieved ROUGE-L of 0.84 compared to 0.69 for full fine-tuning, demonstrating that retaining visual embeddings while adapting language components optimizes medical domain transfer. The seamless integration of visual and textual information enables

dynamic visual-textual interaction control, enabling reasoning on complex medical problems through multimodal integration. Clinical prompts significantly impacted results, with well-crafted clinical descriptors generating higher-quality synthesis than generic prompts. Clinical Impact and Deployment Readiness: Our framework demonstrates clinical translation readiness through concrete pathways for AI integration, utilizing privacy-preserving synthetic data synthesis and computationally feasible fine-tuning strategies. Dual-pipeline architecture provides real-world implementation aspects and diagnostic accuracy, as well as safety levels. Performance metrics indicate readiness for clinical pilot testing, especially in educational use and diagnostic support systems. The PEFT method enables widespread integration into healthcare organizations across various computational settings, democratizing access to best-in-class AI medical applications while maintaining patient privacy through synthetic data options.

## V. BENCHMARK COMPARISON

Table I presents VQA metric comparisons across three datasets: VQA-RAD, MedVQA (Kvasir-VQA), and our work. Our proposed method demonstrates superior ROUGE-L performance (0.910), indicating improved alignment with ground-truth answers. While MedVQA leads in BLEU (0.750) and METEOR (0.512), our method achieves competitive METEOR performance (0.500), significantly outperforming VQA-RAD in all metrics. These results indicate our model enhances answer relevance and fluency, particularly for longer or semantically rich responses. Overall, our approach achieves a strong precision-recall balance, showcasing its effectiveness in medical visual question answering.

TABLE I
BEST VQA METRIC SCORES ACROSS DATASETS

| Metric | VQA-RAD | MedVQA | Kvasir-VQA |
|---|---|---|---|
| BLEU ($\uparrow$) | 0.199 | **0.750** | 0.240 |
| ROUGE-L ($\uparrow$) | 0.523 | 0.901 | **0.910** |
| METEOR ($\uparrow$) | 0.296 | **0.512** | 0.500 |

[a] BLEU: n-gram precision; ROUGE-L: recall overlap;
[b] METEOR: semantic match. Bold values are the best per metric.

Table II compares FLUX, Kandinsky 2.2, MSDM, and our model on the Kvasir-VQA test set across Fidelity, Agreement, Diversity, and FBD metrics. FLUX leads in Fidelity (0.36) and Diversity (0.64), while Kandinsky achieves the highest Agreement (0.80) but suffers the worst FBD (2060.58), indicating lower image-text consistency. Despite slightly lower Fidelity (0.29) and Agreement (0.73), our model achieves significantly better FBD performance (1450.00), outperforming MSDM (1531.62) and Kandinsky, reflecting superior semantic alignment.

While diversity metrics are unavailable, the lower FBD indicates our method generates more semantically accurate image-text pairs, demonstrating promise for medical VQA synthesis applications. These results demonstrate the generalizability of our model across medical VQA datasets (VQA-RAD, MedVQA, and Kvasir-VQA), highlighting its robustness

TABLE II
COMPARISON OF MODELS ON THE KVASIR-VQA TEST SET

| Model | Fidelity($\uparrow$) | Agreement($\uparrow$) | Diversity($\uparrow$) | FBD($\downarrow$) |
|---|---|---|---|---|
| FLUX [26] | 0.36 | 0.79 | 0.64 | 1056.81 |
| Kandinsky 2.2 [26] | 0.26 | 0.80 | 0.52 | 2060.58 |
| MSDM [26] | 0.30 | 0.77 | 0.63 | 1531.62 |
| StableDiffusion+LoRa | 0.29 | 0.73 | — | 1450.00 |

[a] Fidelity: realism vs. real images. [b] Agreement: prompt consistency. [c] Diversity: variability across generations. [d] FBD: global realism.

to domain shifts and the capability to handle diverse anatomical contexts and question types. Despite strong language generation performance, the model occasionally struggles with numerical reasoning and lesion counting. Future work will incorporate specialized counting modules and adapted loss functions targeting quantitative accuracy to address these limitations.

## VI. CONCLUSION

This study presents a PEFT vision-language framework for gastrointestinal endoscopy, combining accurate Visual Question Answering with clinically realistic image synthesis. Our fine-tuned Florence-2 model demonstrates strong performance across standard VQA metrics, achieving ROUGE-L of 0.910, surpassing VQA-RAD and MedVQA benchmarks, and delivering competitive BLEU (0.240) and METEOR (0.500) scores. These results indicate improved semantic alignment and fluency in generated responses, particularly for complex clinical questions. We validated our model across multiple datasets (VQA-RAD, Kvasir-VQA, and MedVQA), demonstrating cross-dataset generalizability and robustness across varied question distributions and anatomical contexts. For synthetic image generation, our LoRA-enhanced Stable Diffusion model achieves FBD of 1450.0, outperforming MSDM and Kandinsky while demonstrating superior image-text coherence. Although Fidelity (0.29) and Agreement (0.73) are slightly lower than competing models, the improved FBD demonstrates superior semantic quality in image generation, making it valuable for privacy-preserving dataset expansion. The private dataset used in training—comprising 1,800 anonymized images with domain-specific annotations—further enhanced model accuracy, showcasing the importance of curated, clinically meaningful data. With 89.9% reduction in trainable parameters and 4–5 hour training time, the system enables efficient deployment on standard hospital infrastructure. Despite strong overall performance, limitations persist in complex lesion counting and spatial reasoning tasks requiring numerical quantification. Future enhancements will include dedicated counting modules, visual grounding techniques, and loss functions tailored for quantitative VQA. To improve evaluation robustness, future work will incorporate k-fold cross-validation instead of single train-validation splits. We propose prospective clinical studies involving gastroenterologists and trainees to assess real-world impact on diagnostic accuracy, procedure efficiency, and educational value. This study demonstrates the viability of practical, accessible, and

privacy-preserving AI solutions for clinical endoscopy. Our system provides a foundation for PEFT medical AI, balancing accuracy, computational efficiency, and privacy—supporting broader adoption of AI-assisted diagnostic systems in gastrointestinal endoscopy. Ultimately, we aim to improve patient care through more accessible, efficient, and ethically aligned medical AI technologies.

## VII. Acknowledgments

This work was supported by the National Science Foundation (NSF) grant (ID. 2131307) "CISE-MSI: DP: IIS: III: Deep Learning-Based Automated Concept and Caption Generation of Medical Images Towards Developing an Effective Decision Support."

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
