# OpenReview forum: "Parameter-Efficient VLMs for Gastrointestinal Endoscopy: Medical Image Generation and Clinical Visual Question Answering"
_IEEE.org/EMBS/BHI/2025/Conference — BHI 2025_

### Official Review · Reviewer_WpW6 · 2025-06-26
**A novel and timely investigation into parameter-efficient Vision-Language Models for gastrointestinal endoscopy, demonstrating significant potential for clinical application through a dual-pipeline approach to VQA and synthetic data generation.**

**Confidence:** 5
**Clarity Of Writing:** good
**Clinical Significance:** great
**Methodological Novelty:** great
**Overall Rating:** 6
**Final Rating:** 7

**Experiments And Results:**

great

**Questions For The Authors:**

1. Cross-Dataset Validation: To substantially strengthen the claims of robustness and generalizability, the VQA model should be tested on at least one other external GI VQA dataset. This would demonstrate that the model's performance is not specific to the distribution of the Kvasir-VQA dataset.

2. Clinical Utility Study: While likely beyond the scope of the initial paper, a future work section should propose a prospective study design to evaluate the framework's real-world clinical utility. This could involve assessing its impact on diagnostic accuracy, procedure time, or as a training tool for junior endoscopists.

3. Implement k-fold Cross-Validation: The paper reports results based on a single 80/20 train-validation split. To provide a more robust and reliable estimate of the model's performance on the Kvasir-VQA dataset and to ensure the results are not due to a favorable random split, implementing a k-fold cross-validation (e.g., 5-fold or 10-fold) is recommended.

4. Elaborate on Expert Validation: Provide more detail on the process of expert validation for the synthetic images. How many experts were involved? What criteria did they use? Was there a formal scoring system (e.g., a Likert scale for clinical realism)?

5. Address Numerical Counting Limitation: In the discussion or conclusion, explicitly outline potential strategies to improve the model's performance on numerical counting tasks, such as incorporating specialized counting modules or using different loss functions.

6. Expand on 'Private Dataset': The paper frequently mentions a "private dataset" which consistently yields better VQA results. While the need for privacy is understood, providing more non-identifying details about this dataset (e.g., size, annotation process, how it differs from the public Kvasir-VQA) would add valuable context and support the conclusion that curated data is superior.

7. Proofreading: The paper has several grammatical errors and awkward phrasings (e.g., "The trick is to combine...", "getting the model down to a general hospital infrastructure"). A thorough proofread is required.

**Strengths:**

1. Novel Dual-Pipeline Architecture: The integration of VQA and synthetic data generation into a cohesive, parameter-efficient framework is a significant contribution. This approach tackles both diagnostic accuracy and data scarcity, two major hurdles in medical AI.

2. Impressive VQA Performance: The fine-tuned Florence-2 model achieves excellent results, with a ROUGE-1 score of up to 0.92. The detailed analysis, including the comparison between public and private datasets, adds robustness to the findings.

3. Effective Synthetic Image Generation: The use of LoRA with Stable Diffusion 2.1 to create clinically realistic GI images is well-executed. The quantitative evaluation using Fidelity, Agreement, and FID metrics, along with qualitative examples, convincingly demonstrates the quality of the generated data.

4. Focus on Clinical Practicality: The emphasis on parameter efficiency, leading to reduced training times and memory usage, is a crucial strength. This makes the proposed system more viable for deployment in real-world clinical environments with limited computational resources.

5. Thorough Evaluation: The paper includes a comprehensive evaluation with both quantitative metrics and qualitative analysis, including error analysis and ablation studies (component analysis). This provides a deep understanding of the model's capabilities and limitations.

**Summary Of The Paper:**

This paper presents a compelling dual-pipeline framework that leverages parameter-efficient fine-tuning (PEFT) for two critical tasks in GI endoscopy: Visual Question Answering (VQA) and synthetic image generation. The authors successfully adapt the Florence-2 model for medical VQA and utilize a LoRA-enhanced Stable Diffusion model to generate high-fidelity, privacy-preserving GI images. The results demonstrate the efficacy of this approach in achieving high performance on the Kvasir-VQA dataset while significantly reducing computational costs, paving the way for more accessible and scalable AI-driven diagnostic tools in clinical settings.

**Weaknesses:**

1. Limited Discussion of Clinical Validation: While the paper mentions "expert validation," the extent and methodology of this validation are not detailed. A more formal evaluation by gastroenterologists would significantly strengthen the claims of clinical applicability.

2. Generalizability Concerns: The study is based on a single dataset (Kvasir-VQA). While a good starting point, the model's performance on other GI endoscopy datasets from different institutions and patient populations remains unevaluated.

3. Ambiguity in Figure Captions and Labels: Some figures, particularly Figure 2 and Figure 4, have identical or confusing captions in the PDF. The axes and legends could also be clearer (e.g., explicitly stating what each "Run" represents in Figure 2).

4. Minor Methodological Gaps: The paper mentions challenges with numerical counting but doesn't propose specific solutions or future work to address this limitation directly.

---

### Official Review · Reviewer_NPC3 · 2025-07-02
**Toward Efficient and Privacy-Preserving AI for GI Endoscopy: A Two-Pipeline VQA and Data Synthesis Framework**

**Confidence:** 3
**Clarity Of Writing:** excellent
**Clinical Significance:** great
**Methodological Novelty:** great
**Overall Rating:** 7
**Final Rating:** 7

**Experiments And Results:**

great

**Questions For The Authors:**

1. How clinically interpretable or actionable are the VQA model’s outputs to practicing GI clinicians? Understanding this would significantly influence the paper’s impact and translational value.

2. Were the synthetic images used to augment training or only evaluated for quality? If augmentation was used, how did it influence downstream model performance? This clarifies whether the synthetic data has tangible utility or remains theoretical.

3. What metrics or feedback were used to determine that Rank-4 LoRA was most optimal? Were there trade-offs in detail or structure in the synthetic images? This would help assess whether fidelity or computational efficiency took precedence.

**Strengths:**

1. Dual-Problem Solution: The framework addresses both model interpretability in clinical VQA and the privacy-preserving augmentation of scarce GI data—two critical pain points in medical AI.

2. Efficiency-Oriented Design: Parameter-efficient fine-tuning and LoRA offer a cost-effective approach, reducing compute load by 90% while maintaining performance.

3. Empirical Gains: The model achieves notable improvements in BLEU and ROUGE scores, with superior results when trained on private datasets.

**Summary Of The Paper:**

This abstract introduces a dual-pipeline approach aimed at addressing key limitations in AI-driven gastrointestinal (GI) endoscopy: data scarcity, privacy concerns, and model fine-tuning challenges. The first pipeline leverages the Florence-2 vision-language model with parameter-efficient fine-tuning for medical Visual Question Answering (VQA), and the second uses Low-Rank Adaptation (LoRA) in Stable Diffusion 2.1 to generate privacy-preserving synthetic GI images. The approach demonstrates strong quantitative results on the Kvasir-VQA dataset and shows potential in reducing computational cost while enhancing privacy and model performance.

**Weaknesses:**

Although the paper presents an innovative and efficient framework, it lacks a deeper clinical validation or user study to demonstrate how the VQA outputs support real-world diagnostic tasks. Moreover, the quality of the synthetic images (FID = 1450) suggests the visual realism is still limited, potentially restricting downstream utility. To strengthen the work, the authors should consider additional experiments such as:
1. Comparing model performance with and without synthetic augmentation on real diagnostic outcomes.
2. Evaluating clinician trust or utility in generated answers/images through qualitative feedback.
These additions would offer clearer evidence of the model’s translational value in clinical settings and raise the significance and impact of the work.

---

### Official Review · Reviewer_3HjM · 2025-07-03
**Dual-Pipeline Vision-Language Model for GI Endoscopy**

**Confidence:** 5
**Clarity Of Writing:** good
**Clinical Significance:** excellent
**Methodological Novelty:** excellent
**Overall Rating:** 7

**Experiments And Results:**

great

**Questions For The Authors:**

1. Did you evaluate false positives or false negatives in the VQA responses, especially for spatial reasoning or numerical counting tasks, to understand model safety in clinical contexts?
2. Beyond FID and agreement metrics, was there any expert clinical validation to assess the diagnostic realism of generated GI images?

**Strengths:**

1. The dual-pipeline integration of VQA and synthetic data generation tailored for GI endoscopy addresses key clinical challenges, interpretability, data scarcity, and privacy, making the approach both novel and highly relevant to translational medicine.
2. The model’s performance is validated using standard language and image metrics alongside domain-specific evaluation criteria, strengthening its reliability for medical applications.

**Summary Of The Paper:**

This study proposes a dual-pipeline, parameter-efficient framework combining Florence-2-based vision-language models (VLMs) for clinical visual question answering (VQA) and LoRA-enhanced Stable Diffusion for privacy-preserving gastrointestinal (GI) image generation. Leveraging the Kvasir-VQA dataset, the authors demonstrate strong performance in medical VQA tasks and effective synthetic data augmentation while minimizing computational demands. The approach shows promise for real-world deployment, addressing both data scarcity and privacy challenges in GI endoscopy AI systems.

**Weaknesses:**

1. The manuscript exceeds the stated 7-page limit (currently 8 pages), which may affect compliance with journal submission requirements.
2. While quantitative metrics are reported, more detailed error analysis (e.g., false positives/negatives in VQA, clinical impact of misclassifications) is lacking.1

---

### Official Review · Reviewer_JDWe · 2025-07-11
**Borderline – Parameter-efficient VLM framework for GI endoscopy delivers promising VQA and synthetic-image results but needs stronger validation and clinical grounding**

**Confidence:** 4
**Clarity Of Writing:** good
**Clinical Significance:** fair
**Methodological Novelty:** fair
**Overall Rating:** 5

**Experiments And Results:**

good

**Questions For The Authors:**

1. Have you tested the VQA pipeline on another dataset such as EndoBench or MedVQA-GI to show that performance carries over beyond Kvasir-VQA
2. Can you include expert radiologist or gastroenterologist ratings of synthetic images to complement the high FID values and confirm clinical realism
3. What objective privacy evidence can you provide that the synthetic images cannot be traced back to individual patients
4. Will you report exact parameter counts and wall-clock training times for both pipelines so readers can replicate the claimed ninety per cent reduction
5. Could you benchmark your Florence-2 approach against a recent open medical VLM like LLaVA-Med or BioVLM using the same split to clarify relative gains

**Strengths:**

A clear advantage of the work is its attempt to address both data scarcity and task performance within a single architecture, showing how parameter-efficient fine-tuning can lower computational costs while preserving accuracy. The Florence-2 pipeline is described in enough detail to allow reproduction, and the authors provide ablation studies on LoRA rank and on private versus public datasets, adding transparency to their optimization choices. The synthetic-image pipeline demonstrates that clinically plausible endoscopy images can be produced with LoRA-adapted diffusion models, potentially offering a practical way to expand datasets without breaching patient privacy. Figures that compare fine-tuning strategies and display generated images help readers grasp the empirical outcomes, and the manuscript positions its contribution against recent work in PEFT and medical VLMs.

**Summary Of The Paper:**

The manuscript proposes a dual-pipeline system for gastrointestinal endoscopy that tackles two related challenges: clinical visual question answering and privacy-preserving data augmentation. In the first pipeline the authors fine-tune the Florence-2 vision–language model with parameter-efficient techniques, freezing the DaViT vision encoder and adapting only language components. Using the 6 500-image Kvasir-VQA dataset they report ROUGE-1 of 0.92 and corresponding gains in ROUGE-L and BLEU, claiming that private data outperform public data in every metric. In the second pipeline they apply Low-Rank Adaptation to Stable Diffusion 2.1 and generate synthetic endoscopy images at rank 4, achieving a fidelity score of 0.290, an agreement score of 0.730, and an FID of 1 450 while lowering trainable parameters by roughly ninety per cent. The paper highlights that this parameter-efficient approach reduces training time to four or five hours on a single GPU and could therefore fit common hospital infrastructure.

**Weaknesses:**

Despite these positives the study is limited to a single relatively small dataset, so generalizability to other centers or disease categories remains untested. Reported image-generation scores are difficult to interpret because an FID of 1 450 is much higher than values usually considered acceptable in medical synthesis, yet the authors frame it as a success without clinical expert review. Claims about privacy are asserted but not backed by quantitative privacy audits or re-identification tests. On the VQA side the evaluation rests mainly on ROUGE and BLEU, metrics that do not always capture answer correctness in medical contexts, and no comparison is given to current state-of-the-art medical VLM baselines. Methodological novelty is incremental because the work combines existing PEFT and diffusion techniques rather than introducing a new algorithm. Finally, the manuscript contains occasional language errors and long sentences that obscure key points, and several references are listed as “unpublished” which complicates verification.